# When, How, and to What Extent Are Individuals with Unresponsive Wakefulness Syndrome Able to Progress? Functional Independence

**DOI:** 10.3390/brainsci10120990

**Published:** 2020-12-16

**Authors:** José Olaya, Enrique Noé, María Dolores Navarro, Myrtha O’Valle, Carolina Colomer, Belén Moliner, Camilla Ippoliti, Joan Ferri, Anny Maza, Roberto Llorens

**Affiliations:** 1NEURORHB, Servicio de Neurorrehabilitación de Hospitales Vithas, Fundación Vithas, Callosa d’En Sarrià 12, 46007 València, Spain; olaya@neurorhb.com (J.O.); loles@neurorhb.com (M.D.N.); myrtha@neurorhb.com (M.O.); carol@neurorhb.com (C.C.); belen@neurorhb.com (B.M.); camila@neurorhb.com (C.I.); joan@neurorhb.com (J.F.); or rllorens@i3b.upv.es (R.L.); 2Neurorehabilitation and Brain Research Group, Instituto de Investigación e Innovación en Bioingeniería, Universitat Politècnica de València, Camino de Vera s/n, 46011 València, Spain; amaza@i3b.upv.es

**Keywords:** unresponsive wakefulness syndrome, vegetative state, minimally conscious state, disorders of consciousness, brain damage, disability, independence, functionality

## Abstract

Accurate estimation of the functional independence of patients with unresponsive wakefulness syndrome (UWS) is essential to adjust family and clinical expectations and plan long-term necessary resources. Although different studies have described the clinical course of these patients, they have methodological limitations that could restrict generalization of the results. This study investigates the neurobehavioral progress of 100 patients with UWS consecutively admitted to a neurorehabilitation center using systematic weekly assessments based on standardized measures, and the functional independence staging of those patients who emerged from a minimally conscious state (MCS) during the first year post-emergence. Our results showed that one year after emergence, most patients were severely dependent, although some of them showed extreme or moderate severity. Clinically meaningful functional improvement was less likely to occur in cognitively-demanding activities, such as activities of daily living and executive function. Consequently, the use of specific and staging functional independence measures, with domain-specific evaluations, are recommended to detect the functional changes that might be expected in these patients. The information provided by these instruments, together with that obtained from repeated assessments of the preserved consciousness with standardized instruments, could help clinicians to adjust expectations and plan necessary resources for this population.

## 1. Introduction

Since the description of vegetative state by Jennet and Plum back in 1972 [1] until well into this century, the clinical view of patients with disorders of consciousness (DOC) was dominated by a unitary concept of consciousness, aligned with many of the theories of functional segregation that sought to explain the nature of cerebral functionality at that time. Much of that view contributed to the understanding of recovery of consciousness as a magical or miraculous event, which has been fed by the treatment of the cases in the media [2]. Consciousness has traditionally been considered a dichotomist entity, according to which patients could be classified as being either conscious or unconscious/vegetative, based on the presence or absence of observable clinical signs of interaction with the environment, respectively. This classification could have modulated the therapeutic effort invested in unresponsive patients, with those patients with worse clinical diagnosis receiving less treatment. According to this effect, diagnosis could become a self-fulfilling prophecy, which entails serious clinical and ethical issues [3]. The description of the minimally conscious state (MCS) by the Aspen Neurobehavioral Conference Workgroup [4] and the subsequent definition of the behavioral diagnostic criteria in 2002 [5], meant a conceptual revolution [6]. The definition of MCS was opposite to the categorical conception of consciousness and supported the idea of recovery of consciousness as a gradual process. The progressive clinical course of patients with DOCs fitted better with the theories of functional integration that emerged by the end of the last century, in light of the new findings on brain connectivity [7]. At the same time, in an attempt to find a differential term to refer to unresponsive patients while avoiding the negative connotations of the term vegetative, the European Task Force on Disorders of Consciousness coined the term unresponsive wakefulness syndrome (UWS) [8].

As our understanding of the neural basis of consciousness and the clinical management of the medical complications associated with patients with DOC improve, an increasing number of studies has documented recovery of consciousness even years after the onset [9,10,11,12,13,14,15,16,17]. Most studies have investigated the level of care dependence and disability of patients with DOC using standardized measures, primarily the Glasgow Outcome Scale [18] or the Disability Rating Scale [19]. The Glasgow Outcome Scale is a global scale that classifies the functional outcome of the patients into one of five categories (or eight, in the case of the extended version of the scale). The Disability Rating Scale is an eight-item scale that addresses general function by assessing impairment, disability and handicap. Giacino and Kalmar examined the clinical progress of a group of 55 patients with UWS over the course of a year [5]. Although the authors found an improvement over time in the Disability Rating Scale, most patients were in a vegetative state or had an extremely severe disability according to this scale one year after inclusion in the study. Katz et al. investigated the long-term outcome of a combined group of 11 patients with UWS and 24 patients in MCS with the same instrument [20]. The available data from seven patients with UWS at a one-year follow-up showed a severe to extremely severe disability. Lauté et al. explored the clinical course of 11 patients with UWS over five years after injury with the Glasgow Outcome Scale, and reported that, at the end of the analyzed period, nine patients with UWS had died and two remained unresponsive [21]. Estraneo et al. investigated the functional outcome of 50 patients with UWS over a variable time period of one to four years and focused on those patients with a late recovery, defined as occurring after one year [22]. At the end of the study, late improvement was detected in 10 patients. Four of them transitioned to MCS and showed an extremely severe disability, and the remaining six patients emerged from MCS and showed a severe to extremely severe disability, according to the Disability Rating Scale. The same authors used this scale to examine the outcome of a group of 43 patients with prolonged UWS caused by an anoxia, two years after the onset [23]. Seven patients transitioned to MCS and were characterized as having either an extremely severe disability or being vegetative. Two patients emerged from MCS and exhibited a severe and an extremely severe disability. Steppacher et al. followed the clinical progress of 59 patients with UWS during a mean period of 8.8 years from the onset with the Glasgow Outcome Scale [24]. At follow-up, 28 patients had died, 19 patients remained vegetative, 11 patients had a severe disability, and one patient had a moderate disability.

Although the Glasgow Outcome Scale and the Disability Rating Scale are standardized clinical instruments, easy to administer, and not time-consuming, they could lack the specificity and sensitivity to detect subtle functional changes, especially in patients with severe and prolonged disability over time, as patients with UWS are [20]. Only four studies to date, three of them describing the clinical progress over the years of the same group of patients, have used a specific independence measure in this population, namely the Functional Independence Measure [25]. This instrument incorporates 18 items that measure independence in activities related to self-care, mobility, and cognition, and allows estimating functional independence in motor and cognitive tasks, separately. In their study, Estraneo et al. followed up 13 previously reported patients with UWS [22,23] who showed a late improvement of their neurobehavioral condition, for a minimum period of five years after the onset [26]. In the final assessment, four patients were in MCS and showed variable disability that ranged from extremely severe to an extreme vegetative state, according to the Disability Rating Scale, and almost had minimal scores in both the motor and cognitive subscales of the Functional Independence Measure. Six patients emerged from MCS, mostly showed a moderately severe disability according to the Disability Rating Scale, and exhibited a variable independence according to the Functional Independence Measure, with scores that ranged from 37 to 65 from a maximum possible score of 91. In a series of three studies, the course of recovery of 110 patients unable to follow commands, who were admitted to inpatient rehabilitation in the Traumatic Brain Injury Model System Programs of the National Institute on Disability and Rehabilitation Research, were examined [27,28,29]. More than half of the patients were functionally independent according to the Functional Independence Measure at a 10-year follow-up. However, this percentage was dramatically lower for those patients who did not emerge in the first 28 days after the injury [29].

Although some valuable reports have described the functional outcome of patients with UWS, they incorporate case studies or small samples [9,10,11,12,13,14,15,16,17,20,21]; include infrequent assessments [5,20,21,22,24,26,27,28,29]; do not use standardized neurobehavioral assessment measures [21,24,27,28,29], contrary to updated recommendations [30,31]; use global measures with limited sensitivity [5,20,21,22,23,24] or non-staging independence measures [26,27,28,29]; focus on patients with late recovery [22,26] or specific etiologies [23,27,28,29]; or combine data with those from patients with MCS [27,28,29]. These limitations might hinder accurate estimation of the achievable independence in patients with UWS, which might be essential to adjust expectations and plan necessary resources for this population.

The objective of this study was, therefore, to investigate the neurobehavioral progress of a representative cohort of patients with UWS who were provided multidisciplinary rehabilitation through systematic weekly assessments based on standardized measures, and the functional independence staging of those patients who emerged from MCS during the first year after emergence.

## 2. Materials and Methods

### 2.1. Participants

A retrospective analysis of the demographic and clinical data of all patients with a brain injury admitted to the inpatient neurorehabilitation program of a network of four hospitals, from January 2004 to January 2020, was conducted. Patients were included if they were admitted to the recruitment centers with UWS after a brain injury of any etiology, according to the Coma Recovery Scale-Revised (CRS-R) [32], if a time between one and 12 months after the injury had elapsed, and if their neurobehavioral condition was clinically monitored for a minimum of 12 months since the injury, emergence from MCS, or decease. Patients whose neurobehavioral condition was not assessed in two consecutive weeks were excluded.

The study was approved by Comité Ético de Investigación Clínica del Hospital Clínic Universitari de València (2019002). Written informed consent to participate in the study was obtained from the legal representatives of all patients.

### 2.2. Procedure

All patients were assessed a minimum of five times using the Spanish version of the CRS-R during the first week after admission [32,33]. The initial diagnosis was made considering the score obtained in each evaluation and the cut-off scores proposed for this scale [32].

Patients were originally included in a multidisciplinary rehabilitation program that included daily sessions of physical therapy and multimodal stimulation customized to their particular needs. The intervention aimed at avoiding and treating possible medical complications derived from the physical condition and the prolonged immobility (i.e., passive exercises to maintain ranges of motion, postural care, daily sitting, etc.), as well as promoting the recovery of consciousness by reducing confounding factors (i.e., agitation or pain), using multisensory stimulation (i.e., interactive projections, aromatherapy, musical selections, tactile stimulation, etc.), drugs (i.e., amantadine, zolpidem, etc.), or non-invasive brain stimulation (i.e., transcranial direct current stimulation or transcutaneous auricular vagus nerve stimulation).

Weekly assessments with the CRS-R were conducted until discharge, decease, or emergence from MCS. In the case of decease, the causes of the death were collected. Patients who emerged from MCS continued the rehabilitation program, which included physical therapy, occupational therapy, cognitive therapy, and speech therapy, adapted to their new clinical situation and functional independence. The functional independence of these patients was assessed six months and one year after the emergence with the Disability Rating Scale [19], the Barthel Index [34], and the Functional Independence Measure [25].

Assessments of the neurobehavioral condition and functional independence were conducted by a trained neuropsychologist during the morning, from 10 a.m. to 12 a.m.

### 2.3. Data Analysis

Interpretation of the independence measures was done separately as the three instruments evaluate different constructs. Scores in the Disability Rating Scale were considered to indicate no disability (score of 0), mild (1), partial (2–3), moderate (4–6), moderately severe (7–11), severe (12–16), extremely severe (17–21), vegetative state (22–24), and extreme vegetative state (25–29) [5]. However, it is important to highlight that these scores have never been validated [35]. Scores in the Barthel Index were considered to indicate total dependence (scores below 21), severe dependence (21–60), moderate dependence (61–90), and slight dependence (scores above 90) [34]. Total score in the Functional Independence Measure was considered as a general measure of functional independence. In addition, the stages of functional independence across the activities of daily living, sphincter management, mobility, and executive function domains were estimated from the scores in the Functional Independence Measure as proposed by Stineman et al. [25]. Stages reflected total assistance (stage of 1), maximal assistance (2), moderate assistance (3), minimal assistance (4), supervision (5), modified independence (6), and complete independence (7). Results in each instrument were analyzed separately, as the three scales have different constructs.

Comparisons of quantitative clinical and demographic variables were performed using paired *t*-tests.

## 3. Results

### 3.1. Participants

A total of 100 patients (29 women and 71 men,) with UWS due to a traumatic (*n* = 40) or non-traumatic cause (*n* = 60) met the inclusion criteria (Figure 1). Patients had a mean age and standard deviation of 37.7 ± 18.0 years, including 14 patients under 18 years of age, and a mean time since injury and standard deviation of 132.8 ± 85.5 days. At admission, 40 patients had a time since injury of less than 3 months, and 77 patients had a time since injury of less than 6 months.

Ten patients died over the period analyzed, either from complicated infectious processes that led to septic conditions, as registered in eight cases, or cardiorespiratory arrest, as registered in the remaining two cases. No deaths were caused by withdrawal of nutritional or hydration support. Mean time from injury to death was 376.7 ± 291.5 days. Seven patients died during the first year after the injury and the remaining three patients died after this period. At the time of death, two patients had progressed to the MCS and had a mean score in the CRS-R of 12.5 ± 0.7. The remaining eight patients were in UWS and had a mean score in the CRS-R of 6.1 ± 1.4 at the time of death.

Twelve patients, 11 men and a woman, progressed to the MCS and then emerged from the MCS. At the moment of emergence, patients had a mean age of 24.2 ± 6.9 years, a mean time since injury of 251.1 ± 160.6 days, and a mean score in the CRS-R of 19.1 ± 2.9. The neurobehavioral progress of each patient is provided in Table 1. Patients who emerged from MCS were younger than those who did not (t(98) = 2.805, *p* = 0.006), were more likely to have suffered a traumatic injury (Χ^2^ (1, *n* = 100) = 10.669, *p* = 0.001), and had higher scores in the CRS-R at admission (t(98) = −2.504, *p* = 0.014). No differences in sex, education, or time post-injury at admission were detected.

Complementary information about the neurobehavioral progress of the participants is provided in an accompanying or sister article [36].

### 3.2. Functional Independence

The general functional independence of the patients improved over time, according to the results of all measures in the 6-month and 12-month assessments (Table 2).

Patients emerged from the MCS with a severe or extremely severe disability, and had a mean score in the *Disability Rating Scale* of 16.2 ± 2.5. After six months, three patients improved to a moderately severe disability and another patient improved to a moderate disability. An additional patient showed a moderately severe disability one year after the emergence from the MCS. The mean scores in the 6-month and 12-month assessments were 14.3 ± 4.1 and 13.2 ± 4.9, respectively. Improvements were statistically significant both from emergence to the 6-month assessment (t(11) = 2.919, *p* = 0.014) and from the 6-month to the 12-month assessment (t(11) = 2.399, *p* = 0.035). The individual progress of each patient is shown in Figure 2.

The Barthel Index identified all patients as being totally dependent at the emergence and showed a mean score of 2.3 ± 3.9. Six months after emerging from the MCS, the level of dependence of half of the patients improved from total to severe, and the mean score in this scale increased to 19.2 ± 20.3. One year after emerging from MCS, five patients were totally dependent, five patients were severely dependent, and the remaining two patients were moderately dependent (and almost reached the threshold of slight dependence). The mean score in the Barthel Index was 31.5 ± 30.1 at the 12-month assessment. Improvements from emergence to the 6-month assessment (t(11) = −3.413, *p* = 0.006) and from the 6-month to the 12-month assessment (t(11) = −3.125, *p* = 0.010) were statistically significant. The individual progress of each patient is shown in Figure 3.

The assessment with the Functional Independence Measure showed that patients improved from a mean total score of 20.2 ± 3.1 at the moment of emergence from MCS to a mean score of 37.2 ± 20.9 six months later (t(11) = −3.295, *p* = 0.007), and continued to increase until 44.0 ± 26.9 at the 12-month assessment (t(11) = −2.688, *p* = 0.021). The individual progress of each patient is shown in Figure 4.

According to the staging of the functional independence, at emergence from the MCS, all the patients required total or maximal assistance in all domains (Table 3, Figure 5). Six months later, one patient was completely independent in sphincter management and required moderate or minimal assistance in other domains. Seven patients were, in contrast, still completely dependent in all domains. The remaining four patients were also completely dependent in all domains but one: one patient was completely independent in sphincter management and two required minimal and moderate assistance in this domain, and one patient was independent in mobility using a device. One year after emergence from MCS, two patients required moderate or minimal assistance in activities of daily living or executive functioning, while the other patients were completely dependent. Three patients were completely independent in sphincter management, three patients needed supervision, two patients required moderate assistance, and the remaining four patients required total assistance in this domain. The same patient who was independent in mobility using a device six months after emergence remained in the same condition. Two patients required minimal or moderate assistance in this domain, and other patients were completely dependent.

## 4. Discussion

The present study describes the neurobehavioral course of a cohort of 100 patients with UWS consecutively admitted to a multidisciplinary rehabilitation unit and the functional independence staging of a subgroup of 12 patients who emerged from MCS during the first year post-emergence. According to our results, during the analyzed period, one tenth of the patients with UWS died, nearly one third of the patients were able to progress from UWS to MCS, and nearly one tenth of the total sample (nearly one third from those who progressed to MCS) were able to emerge from MCS. However, recovery of consciousness was rarely matched by regaining functional independence. Our results indicated that all the patients who emerged still required significant assistance one year after emergence, with cognitively-demanding activities being specially challenging, which supports the use of specific and staging functional independence measures, with domain-specific evaluations. The information provided by these instruments, together with that obtained from repeated assessments of the preserved consciousness with standardized instruments, could help clinicians to adjust expectations and plan necessary resources for this population.

Although the methodological differences with previous studies do not allow for precise comparison, the variable but serious disability evidenced by our results in the Disability Rating Scale, which showed moderate to extremely severe disability one year after emergence from MCS, supports previous findings. Estraneo et al. reported a severe disability after two years post-onset in two anoxic patients who emerged from MCS [23]. The same group of researchers reported severe to extremely severe functional disability after an unknown time from emergence from MCS in a group of six patients with different etiologies and late recovery [22]. Interestingly, this group of patients improved up to a severe disability after a mean time from the emergence of 57.3 ± 9.6 months [26]. When assessed with the Functional Independence Measure in the same examination, the same patients had scores that ranged from 11 to 31 in the cognitive subscale and from 13 to 34 in the motor subscale [26]. These results are also in line with our results, but for the case of patients 10 and 11, who had remarkably higher scores in the motor subscale.

However, it is important to highlight that the time from emergence to the last assessment was one year in our study, and higher than four years in the study by Estraneo and colleagues [26]. The higher timing of the last assessment in the latter study did not reflect improvements in the functional independence of the patients, even though they had much more time to improve. Although the number of patients is not sufficient to establish reliable associations, this could evidence a flattening of the clinical progression. This effect has been recently described in certain recovery patterns, estimated through multi-trajectory modeling of collected data from unresponsive patients of the Traumatic Brain Injury Model System Programs of the National Institute on Disability and Rehabilitation Research [29]. Although this work included all patients who were unable to follow commands and, consequently, distinction between DOCs and other pathological conditions was not possible (and even less between UWS and MCS), it could provide an approximate description of the course of functional recovery after traumatic brain injury. This study also showed a greater difficulty at regaining independence with cognitive tasks. While almost three-quarters of the followed-up patients were independent with self-care and mobility, only one quarter was independent with cognitive tasks 10 years after post-injury [29]. In the study by Estraneo et al. almost all the six patients who emerged from MCS had impaired abstract reasoning, selective attention, and verbal learning more than four years post-injury [26]. In line with this, all patients in our study needed some kind of assistance to perform the activities of daily living and those that involved executive functioning, according to the functional independence staging. In contrast, half of the patients were independent or only needed to be supervised during tasks that required sphincter management. Mobility was also challenging for all our patients, with only two patients, the aforementioned patients 10 and 11, who were minimally dependent and independent with a device, respectively.

The coherence of the estimated disability and level of dependence of these patients across all instruments used in our study highlights the consistency of the measures. Patients 10 and 11 showed the highest functional independence consistently in all the Disability Rating Scale, the Barthel Index, and the Functional Independence Measure. The comparably higher independence of these patients was especially evident in the visual representation of the individual progress of the last two measures. Interestingly, the same six patients with the highest functional independence were ranked in the same order in both the Barthel Index and the Functional Independence Measure. The greater weight given by the Disability Rating Scale to cognitive functioning and the apparent difficulties of patients who emerge from MCS to progress in this domain could limit the sensitivity of this instrument to detect changes in this group of patients. Although the Disability Rating Scale could be useful to track general functional changes over the course of recovery in patients with more severe disability, it could have limited ability to detect changes in those patients who are able to progress in functions less cognitively demanding. Our results could evidence an increased sensitivity of specific independence measures, as the Barthel Index and the Functional Independence Measure, to detect changes in these patients. The separate interpretation of the motor and cognitive abilities and the staging of functional domains of the Functional Independence Measure could improve the accuracy of the estimation of the achievable level of dependence in patients with UWS, the adjustment of expectations and the planning of the necessary resources for this population.

All our findings should be interpreted taking into account the characteristics of our sample. First, all patients included in our study were provided with multidisciplinary rehabilitation in a specialized neurorehabilitation center, which could restrict the generalization of our results to other populations without access to rehabilitation resources. Second, it should be considered that time since injury to admission was variable and was higher than three months in half of our patients. Although these values of time since injury may be representative of patients with UWS at admission to neurorehabilitation centers, it should be taken into account that time post-injury may influence both the percentage of patients who emerge from MCS or die [30,31,37,38]. Third, the assessment of the neurobehavioral condition was exclusively done with the CRS-R, which might have limited sensitivity to detect the ability of patients with impaired motor output to show some degree of consciousness [39]. Although the addition of neurophysiological or neuroimaging examinations in the assessment protocol could have improved the detection of command following [40,41], the restricted access to instrumented tests in comparison to bedside instruments, could have limited the extrapolation of the findings. It is important to highlight that the CRS-R is considered the reference standard (in the absence of a gold standard) of the clinical bedside evaluation for signs of consciousness [30,31]. Finally, the follow-up period after emergence from MCS was limited to one year, which may have prevented identification of further improvements. This could be particularly relevant considering that functional recovery may be slower in these patients [27,28,29]. However, the systematic weekly assessments of the neurobehavioral condition conducted in our study and the estimation of the functional independence using specific independence measures and staging support the reliability of the clinical course detected during the analyzed period. The information provided by these measures could help clinicians to adjust expectations to the overall poor prognosis of this population and plan all necessary resources for them.

## 5. Conclusions

Systematic assessment of the clinical course of 100 patients with UWS with standardized and specific neurobehavioral and functional measures showed that emergence from MCS, detected in 12 patients, was far from indicating complete functional recovery. Conversely, most patients were severely dependent one year post-emergence, especially in cognitively-demanding activities, as activities of daily living and executive function. Global disability measures could be less sensitive to the functional changes that might be expected in these patients. Use of specific and staging functional independence measures, with domain-specific evaluations, are recommended instead to improve the adjustment of clinical and family expectations and the planning of necessary resources for this population.

## Figures and Tables

**Figure 1 brainsci-10-00990-f001:**
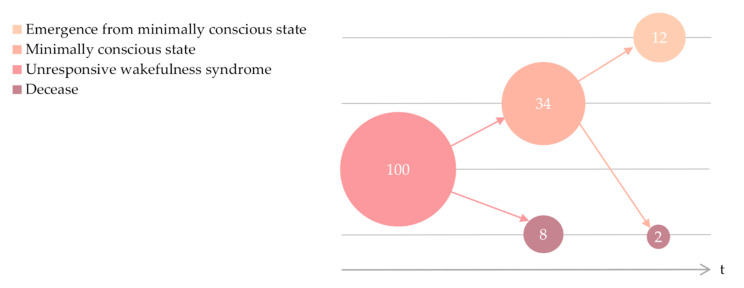
Neurobehavioral progress of the included cohort.

**Figure 2 brainsci-10-00990-f002:**
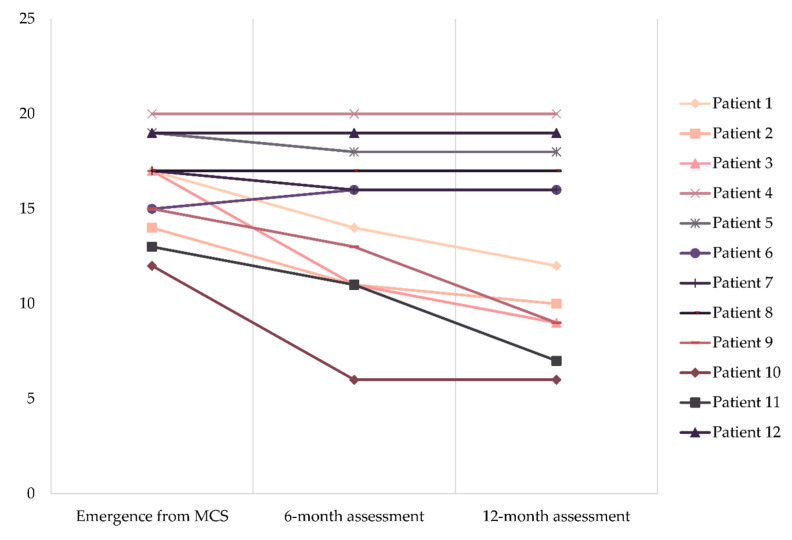
Individual progress in the Disability Rating Scale.

**Figure 3 brainsci-10-00990-f003:**
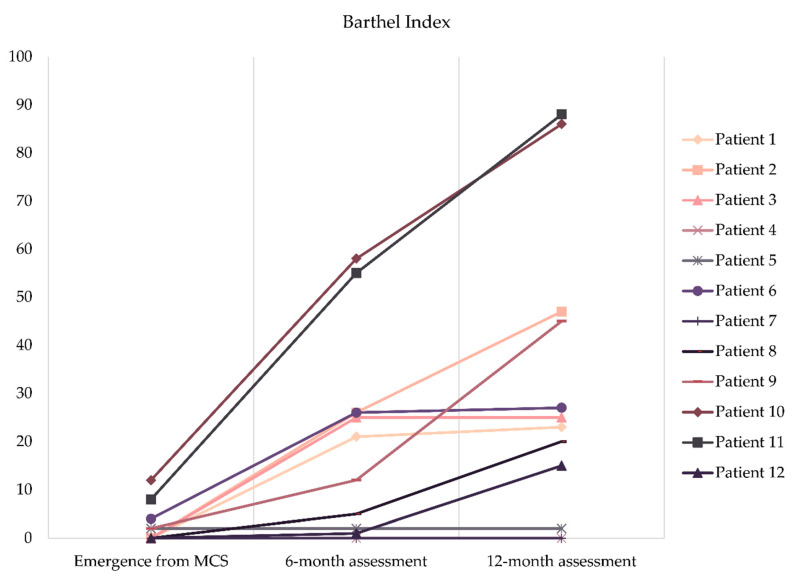
Individual progress in the Barthel Index.

**Figure 4 brainsci-10-00990-f004:**
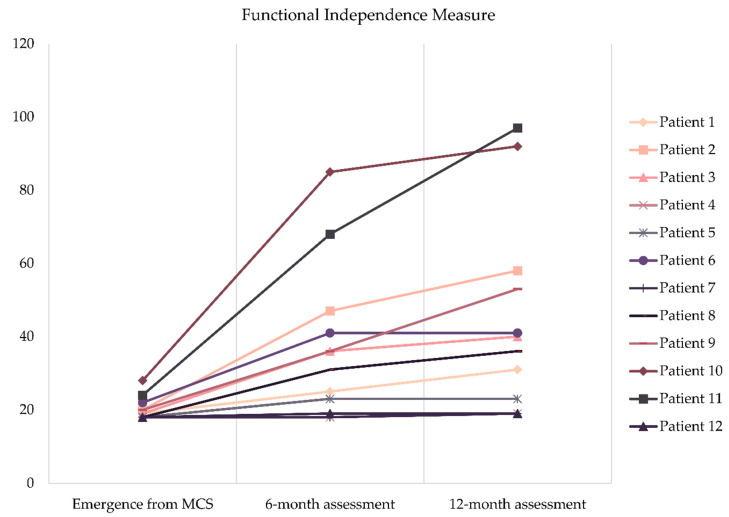
Individual progress in the Functional Independence Measure.

**Figure 5 brainsci-10-00990-f005:**
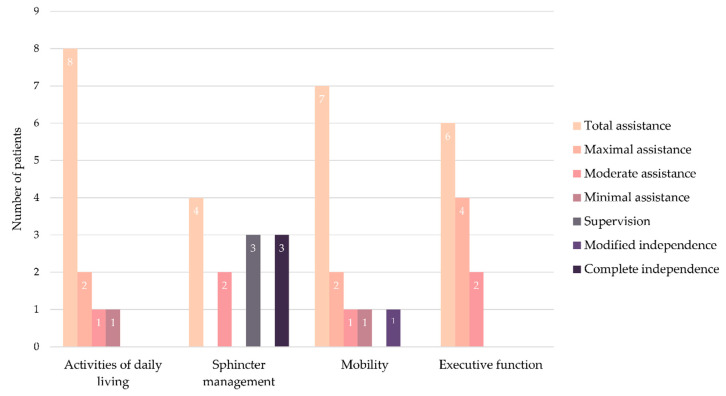
Distribution of patients according to their functional independence staging in each domain one year after emergence from a minimally conscious state.

**Table 1 brainsci-10-00990-t001:** Individual characteristics and neurobehavioral progress until emergence from the minimally conscious state.

Patients	Characteristics		Admission			Progression from UWS to MCS					Emergence from MCS				
	Sex	Etiology	Age	Time Since Injury	CRS-R	Age	Time Since Injury	Time Since Admission	CRS-R	Signs	Age	Time Since Injury	Time since Admission	CRS-R	Signs
Patient 1	M	Traumatic	25	86	5	25	110	24	6	V	26	271	185	20	C
Patient 2	M	Anoxia	14	38	7	14	76	38	10	V	14	104	66	22	C
Patient 3	M	Traumatic	26	38	6	26	76	38	18	M	26	132	94	23	C/FU
Patient 4	M	Traumatic	19	240	3	20	486	246	14	V/M	20	668	428	20	C
Patient 5	W	Traumatic	31	146	7	31	178	32	8	V	32	346	200	19	C/FU
Patient 6	M	Traumatic	31	62	8	31	89	27	10	M	31	172	110	18	C/FU
Patient 7	M	Traumatic	25	213	8	25	250	37	10	V/M	25	401	188	16	C
Patient 8	M	Traumatic	33	45	6	33	91	46	11	V/M	33	201	156	19	FU
Patient 9	M	Fat embolism	19	45	6	19	86	41	10	C	19	99	54	12	C
Patient 10	M	Traumatic	21	117	8	21	181	64	9	V	21	243	126	21	FU
Patient 11	M	Traumatic	13	126	8	13	158	32	12	V	13	222	96	21	FU
Patient 12	M	Traumatic	31	101	7	31	130	29	9	V	31	154	53	18	FU

Age is expressed in years. Time since injury and time since admission are expressed in days. UWS: Unresponsive Wakefulness Syndrome. MCS: Minimally Conscious State. CRS-R: Coma Recovery Scale-Revised. M: man. W: woman. V: visual. C: communication. M: motor. FU: functional use.

**Table 2 brainsci-10-00990-t002:** Individual progress in functional independence.

Patients	Emergence from MCS					6-Month Assessment					12-Month Assessment				
	Disability Rating Scale	Barthel Index	Functional Independence Measure			Disability Rating Scale	Barthel Index	Functional Independence Measure			Disability Rating Scale	Barthel Index	Functional Independence Measure		
			Motor	Cognitive	Total			Motor	Cognitive	Total			Motor	Cognitive	Total
Patient 1	17	0	13	6	19	14	21	14	11	25	12	23	20	11	31
Patient 2	14	0	13	7	20	11	26	30	17	47	10	47	36	22	58
Patient 3	17	0	13	6	19	11	25	24	12	36	9	25	26	14	40
Patient 4	20	0	13	5	18	20	0	13	5	18	20	0	13	6	19
Patient 5	19	2	13	5	18	18	2	14	9	23	18	2	14	9	23
Patient 6	15	4	15	7	22	16	26	29	12	41	16	27	29	12	41
Patient 7	17	0	13	5	18	16	0	13	5	18	16	0	13	6	19
Patient 8	17	0	13	5	18	17	5	24	7	31	17	20	28	8	36
Patient 9	15	2	13	7	20	13	12	21	15	36	9	45	36	17	53
Patient 10	12	12	19	9	28	6	58	62	23	85	6	86	68	24	92
Patient 11	13	8	16	8	24	11	55	52	16	68	7	88	78	19	97
Patient 12	19	0	13	5	18	19	1	13	6	19	19	15	13	6	19

Lower scores in the Disability Rating Scale indicate lower disability. Higher scores in the Barthel Index and the Functional Independence Measure indicate higher independence.

**Table 3 brainsci-10-00990-t003:** Individual progress in the stages of functional independence.

Patients	Emergence from MCS				6-Month Assessment				12-Month Assessment			
	Activities of Daily Living	Sphincter Management	Mobility	Executive Function	Activities of Daily Living	Sphincter Management	Mobility	Executive Function	Activities of Daily Living	Sphincter Management	Mobility	Executive Function
Patient 1	1	1	1	1	1	1	1	1	1	3	1	1
Patient 2	1	1	1	1	1	7	1	2	1	7	1	3
Patient 3	1	1	1	1	1	4	1	1	1	5	1	2
Patient 4	1	1	1	1	1	1	1	1	1	1	1	2
Patient 5	1	1	1	1	1	1	1	1	1	1	2	1
Patient 6	1	1	1	1	1	3	2	1	1	3	2	1
Patient 7	1	1	1	1	1	1	1	1	1	1	1	1
Patient 8	1	1	1	1	1	2	2	1	2	5	3	1
Patient 9	1	1	1	1	1	1	1	2	2	5	1	2
Patient 10	1	1	2	1	4	7	4	3	4	7	4	3
Patient 11	1	1	1	1	1	1	6	2	3	7	6	2
Patient 12	1	1	1	1	1	1	1	1	1	1	1	1

Higher scores indicate higher independence.

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
