# Peer review of "When, How, and to What Extent Are Individuals with Unresponsive Wakefulness Syndrome Able to Progress? Functional Independence"

_brainsci, 2020, doi:10.3390/brainsci10120990_

Round 1

Reviewer 1 Report

J. Olaya and colaborators present the results of retrospective assessment of functional outcomes in a large cohort of UWS patients (n=100). The primary focus of the study was measuring the level of independence in patients emerged from MCS (n=12). Authors state that, despite recovering consciousness, most patients did not achieve independence, and that specific functional independence measures may be more sensitive to functional changes than disability measurements.

There are no serious concerns regarding the implementation of the assessment of functional independence of patients who regained consciousness. The introduction section covers the studies of functional outcome in chronic DOC. Functional Independence Measure is presensted as an instrument that is thought to provide more detailed assessement of independence than more global disability ratings. Description of patient population is accessible; however, for it was not stated whether UWS patient remained hospitalized until they emerged to MCS or they were discharged at some time point and then followed up, with CRS-R assessment continued. In Figure 1 No of patients emerged from MCS is 11 instead of 12, as described in text and tables.

The Results section mostly describe patients emerged from MCS. Measures of disability and Functional Independence Measure demonstrate coherent outcomes, with the latter instrument providing more detailed description of independency domains. Lines 23 and 24 in the Discussion section might be erroneous (one year assesment is said to be "higher than four years" in previous study). In general, results are congruent with previous studies. However, the paper will benefit greatly if the overall significance of the results for clinical practice and their novelty would be explained in more detailed manner.

It should be noted that data regarding neurobehavioral progress of UWS patients was not analyzed in much detail, as it was not apparently the main objective of the paper. Functional outcomes of UWS collected in large longitudinal studies are still of great interest, esepecially dealing with prognostic aspects. Here, track of progression from UWS to MCS is not detailed. E.g., among MCS patients who eventually regained consciousness, the majority were victims of trauma; transition from UWS to MCS was recorded after median of 37.5 days since admission/120 days since injury - was this associated with natural course of DOC, better diagnosis, treatment of medical complications of rehabiliatiion program? The article does not discuss these issues, however, publication of these analyses might be interesting.

Reviewer 2 Report

This manuscript describes an interesting longitudinal study of the functional recovery of patients in prolonged UWS, admitted to a rehabilitation hospital. The study set-up seems sound, although some details regarding data acquisition are lacking. The results on the EMCS patients are interesting, however, the authors could increase efforts to retrospectively investigate if there is any specific characteristic to these recovered patients. Furthermore, I believe the valuable data of the patients who did not recover to EMCS merit further exploration.

Introduction

line 60: Although authors à although the authors

line 88: at al. à et al.

Methods

Line 122: “Patients whose neurobehavioral condition was not assessed in two consecutive weeks were excluded.” It is unclear what is meant here, are the patients assessed once every 2 weeks over the course of the study? Could be rephrased for clarity.

Were the CRS-R assessments performed by trained neuropsychologists? Were the CRS-R assessments always performed in the same time of the day, or did that change?

Line 148: For the reader not very familiar with the disability rating scales, it could be useful to briefly mention some hallmark recovery points within the different independence measures. E.g. what can a patient with moderate dependence or assistance do alone, and what not?

Results

At which time point the mean CRS-R scores of the UWS patients was 6.1 (line 175)?

Line 176-197: Did the patients who recovered have a different age/ time since injury than the patients who did not recover?

After reading the introduction and methods section, it was not clear to me that only data from patients who transition to EMCS was reported in this study. In my opinion it is of interest to also report results about the patients who remained in UWS (e.g. did they always show the same reflex behaviors, or did it change over the weeks?), and MCS patients (e.g. did they show the same or different signs of consciousness over the weeks, did their CRS-R scores fluctuate? Were they MCS consistently or did some assessment suggest these patients were UWS again?). Do the consider this information not relevant? In my opinion the closer exploration of the characteristics of patients who remain UWS/MCS would be a valuable addition to the current paper.

The y-axis of Figure 4. is unclear to me. The functional independence measure consists of 7 stages according to the methods section of the manuscript. The y-axis ranges from 0-120. Is the data transformed?

The column names of the tables are sometimes cut in strange places. This could be improved for esthetic reasons.

Did the patients who recovered to EMCS benefit from a specific treatment more often than the other patients? It would be interesting to know if e.g. tDCS or any other therapy part of the rehabilitation program seems to be beneficial for long term recovery, even if at the group level.

Discussion

Perhaps the authors could complement the discussion with some recommendations for the long-term follow-up and assessment of patients with severe consciousness disorders.

Round 2

Reviewer 2 Report

I would like to thank the authors for the clarifications.

Regarding the parallel paper about the UWS and MCS patients, I would suggest to refer to that work, perhaps as in Press, or if already accepted then as a normal citation.

I find the demographic/ treatment information about recovering EMCS patients as compared to the patients who did not recover valuable and would suggest to provide this information in the manuscript.

In the results:

"Patients who emerged from MCSwere 15 years younger, had a double proportion of traumatic injuries, and one-point increased score in the CRS-R at admission. However, there were no differences in their pharmacological treatment (Amantadine, Zolpidem, etc. according to their responses) or rehabilitation protocols."

In the discussion (in slightly modified form):

"Specifically, regarding tDCS (and other non-invasive brain stimulation techniques), as it is relatively recent technique (we could argue about this),we still do not have enough records of patients to establish a proper analysis on the influence of this interventions on long-term functional independence. However, our preliminary exploration did not exhibited differences in the probability of patients who emerge with time. Other databases, including more patients who were admitted before the 2000s, could show different results. A multicenter collaboration could help to explore this hypothesis in the future."
